# Educational patterns of health behaviors and body mass index: A longitudinal multiple correspondence analysis of a middle-aged general population, 2007–2016

**Ana Silvia Ibarra-Sanchez**[1]*, **Birgit Abelsen**[1], **Gang Chen**[2], **Torbjørn Wisløff**[3]

**1** Department of Community Medicine, UiT The Arctic University of Norway, Tromsø, Norway, **2** Centre for Health Economics, Monash University, Melbourne, Australia, **3** Health Services Research Unit, Akershus University Hospital, Lørenskog, Norway

* ana.s.sanchez@uit.no

**Data Availability Statement:** It is not possible to share our data due to the potential of reverse identification of de-identified sensitive participant

## Abstract

Social differences in body mass index and health behaviors are a major public health challenge. The uneven distribution of unhealthy body mass index and of unhealthy behaviors such as smoking, physical inactivity, and harmful alcohol consumption has been shown to mediate social inequalities in chronic diseases. While differential exposures to these health variables have been investigated, the extent to which they vary over the lifetime in the same population and their relationship with level of education is not well understood. This study examines patterns of body mass index and multiple health behaviors (smoking, physical activity and alcohol consumption), and investigates their association with education level among adults living in Northern Norway. It presents findings from a longitudinal multiple correspondence analysis of the Tromsø Study. Longitudinal data from 8,906 adults aged 32–87 in 2007–2008, with repeated measurements in 2015–2016 were retrieved from the survey's sixth and seventh waves. The findings suggest that most in the study population remained in the same categories of body mass index and the three health behaviors at the follow-up, with a clear educational gradient in healthy patterns. That is, both healthy changes and maintained healthy categories were associated with the highest education levels. Estimating differential exposures to mediators of health inequalities could benefit policy priority setting for tackling inequalities in health.

## Introduction

Social differences in health persist and are growing markedly, even in increasingly affluent countries with welfare states [1–3]. Chronic diseases account for the largest part of the social gradient in life expectancy and total mortality [4–6]. Smoking, harmful alcohol consumption, physical inactivity, poor diet and high body mass index (BMI) increase the risk of developing chronic disease [7, 8] and are also unequally distributed across socioeconomic groups [9]. Monitoring social inequalities in the burden of chronic diseases and their determinants can help in developing policies to improve health equality.

information. The data can however be made available upon request to the Tromsø Study once applying for data access. Contact information for the Tromsø study can be found in the following link: https://uit.no/research/tromsostudy/project?pid=709148. The applications are handled by the Tromsø Study Data and Publication Committee. The authors of this study are not made responsible for ensuring access to data from the Tromsø Study.

**Funding:** The author(s) received no specific funding for this work.

**Competing interests:** The authors have declared that no competing interests exist.

In Norway, absolute and relative inequalities in all-cause mortality between education groups are among the largest in Europe [10]. Women and men with the highest education levels live five to six years longer and have better health than those with the lowest education levels [11]. In addition, large socioeconomic inequalities in high BMI and single health behaviors have been observed [12]. Smoking, physical inactivity, alcohol dependency, lower fruit and vegetables consumption are more common among people with lower socioeconomic conditions [11, 13–18].

Although there is extensive research exploring social inequalities in BMI and in individual health behaviors, less is known about social differentials in multiple health behaviors and BMI within the same study cohort. Moreover, there is a knowledge gap in the extent to which these variables vary over time in the same population and how these patterns relate to educational attainment.

First, it is important to address many health behaviors together with BMI, due to the increased risk of chronic diseases and all-cause mortality associated with a higher number of unhealthy modifiable risk factors [8]. Second, following the same individuals over an extended period conveys a broader picture of the long-term exposure effects on the outcome of interest, thereby making it possible to understand the underlying causes of trends or systematic patterns over time.

Previous studies on health behavior trends and their association with diverse social categories have reported contrasting findings. A repeated cross-sectional study from the United States reported the tendency of health behaviors to cluster and persist over time. In this study, the largest group at each time point was comprised of individuals who neither consume fruit and vegetables nor engage in risky behaviors such as smoking and drinking. This study found that males and, in general, participants with low income and education levels were more likely to be in this group [19]. A longitudinal study that followed British men over an extended period found that unhealthy behaviors such as smoking, physical inactivity and high alcohol consumption were strongly associated with low socioeconomic status, and these associations remained over time [20]. A recent longitudinal study using repeated cross-sectional data from Germany found educational variation in BMI and multiple health behaviors, both separately and collectively [21]. Studies on Scandinavian populations that addressed more than two health behaviors found educational inequalities in social participation [22] and motivation to increase physical activity [23], in addition to smoking and physical activity. Additional empirical contributions to health behavior dynamics and their relationship with socioeconomic status over time have shown that different indicators of socioeconomic position may shape health behavior over people's lifetime through different pathways [24, 25]. However, observations from longitudinal studies have suggested that a high percentage of individuals follow a pattern of long-term adherence to the same health behaviors [20] and to the same BMI category [26]. Longitudinal studies that follow BMI and multiple health behaviors in the same study sample are scarce, and this study adds to the literature by investigating social inequality in BMI and health behaviors with longitudinal data that include both men and women. Therefore, this paper aims to research the relationship between the patterns of BMI and three health behaviors (smoking, physical activity and alcohol consumption) and education level using longitudinal data from a population-based health survey of people living in Tromsø, Norway.

## Materials and methods

### Population study and sample

The Tromsø Study is a prospective cohort of residents of the municipality of Tromsø in Northern Norway, which has about 80,000 inhabitants. The study consists of seven surveys (Tromsø

1–7) conducted from 1974 to 2016 with representative samples of the population [27]. A total of 12,984 men and women aged 30–87 participated in Tromsø 6 (2007–2008), and 21,083 men and women aged 40–99 participated in Tromsø 7 (2015–2016). By the sixth wave of the Tromsø Study, data on health behavior were standardized. To study BMI and health behavior dynamics in the same population, eligible participants for this longitudinal study were those who participated in both Tromsø 6 and 7 (N = 8,906). The characteristics of the participants of Tromsø 6, Tromsø 7, and this cohort sample are presented in S1 Table.

The study was approved by the regional committee for Medical and Health Research Ethics (ID: REK 2019/607). Informed consents were obtained from all study participants. In addition, consent for future usage of data for research purpose was obtained.

## Variables

This study focuses on BMI and three health behaviors (smoking, physical activity and alcohol consumption). The variable categories for BMI and the three health behaviors were coded to fit health recommendations. That is, to avoid smoking and high alcohol consumption (more than 14 units per week for men and seven units per week for women), engage in physical activity for at least 150 minutes per week and maintain a normal BMI (18.5–24.9 kg/m$^2$) [28–32].

## Smoking

Participants' smoking status was obtained from the question: "Do/did you smoke daily? a) Yes, now b) Yes, previously c) Never". A variable was coded to represent these three possible answers to this question.

## Alcohol consumption

A variable of alcohol consumption in units per week was created based on questions concerning frequency and units of consumption. The responses to both questions were converted into numerical values to estimate the units per week (units per week = units × frequency). The answers to these questions were harmonized by the survey as follows: 1) "How often do you usually drink alcohol?" a) Never = 0, b) Monthly or less frequently = 0.25, c) Two to four times a month = 0.75, d) Two to three times a week = 2.5, and e) Four or more times a week = 5.5. 2) "How many units of alcohol (one beer, glass of wine, or other beverage) do you usually drink when you consume alcohol?" a) One to two = 1.5, b) Three to four = 3.5, c) Five to six = 5.5, d) Seven to nine = 8 and e) Ten or more = 12. The cut-off point for high alcohol consumption was more than fourteen units per week for men and more than seven units per week for women, as recommended by current health guidelines [29, 30].

## Physical activity

A variable indicating the amount of physical activity in minutes per week was created based on questions regarding frequency and duration (minutes per week = duration × frequency). The answers to these questions were harmonized by the survey as follows: 1) "How often do you exercise (i.e., walking, skiing, swimming, or training any sports)?" a) Never = 0, b) Less than once a week = 0.5, c) Once a week = 1, d) Two to three times per week = 2.5, and e) Approximately every day = 5. 2) "On average, how long do you exercise for?" a) Less than fifteen minutes = 10, b) Fifteen to twenty-nine minutes = 22, c) Thirty to sixty minutes = 45, d) More than one hour = 90. Respondents were classified as having either less than 150 minutes or 150 or more minutes of physical activity per week as recommended by current health guidelines [30–32].

## Body Mass Index (BMI)

BMI was calculated using the objective measure of the participant's height and weight (BMI = weight [kg] / height$^2$ [m$^2$]). Respondents were classified according to standard BMI classification: underweight (under 18.5 kg/m$^2$), normal weight (18.5 to under 25 kg/m$^2$), overweight (25 to under 30 kg/m$^2$) and obese (30 kg/m$^2$ and over) [33].

## Education

Education levels were ascertained from the question: "What is the highest education level you have completed? a) Primary/partly secondary education (up to 10 years of schooling), b) Upper secondary education (minimum of three years), c) Tertiary education, short: college/ university, less than four years, d) Tertiary education, long: college/university, four years or more."

## Statistical analysis

Multiple correspondence analysis (MCA) is a multivariate statistical method of dimension reduction that has become one of the standard tools for interpreting survey data in the social sciences [34]. It is applied to obtain a spatial map of the data's significant dimensions, where proximities between points and the map's other geometric features indicate associations between dimensions [35]. This method reveals the data's main structures, such as the patterns of correlations between variables or similarities between the observations within complex datasets [36]. In MCA, a multi-way contingency table is transformed into an indicator matrix or a Burt matrix and then the algorithm of correspondence analysis is applied [37]. Since MCA is a plot of the chi-square distances of dimensions, the plot can be regarded as a visualization of the chi-square test when taking more than two variables into account. The plot can be seen as a way of reporting variability, rather than testing whether p-values are below a certain pre-specified value [38]. An additional advantage of this method is that there is no need to meet assumptions requirements [39, 40].

Thirty-three variables were created to represent the possible changes in each participant's BMI and health behavior categories, including those categories that remained unchanged at the time of the follow-up. The solution space of was constructed by excluding participants with missing data and categories with a very low count (less than 1%), as recommended by Jones and colleagues [20]. To study the relationship with socioeconomic position, education level was included as a supplementary variable. Supplementary points define additional profiles that are not used to establish the solution space but are projected onto the space afterwards [36]. Analyses stratified by sex and age were performed to account for confounding in the relationship between education and health behavior. The age groups were chosen based on Norway's 1959 education reform, which made seven years of primary education mandatory. Thus, two age groups were created (age 32–47 and 48–87). All analyses were performed using R version 4.1.1.

## Results

Daily smoking decreased notably between the baseline and follow-up, and while the prevalence of low physical activity also decreased, high alcohol consumption and obesity increased (S1 Table). A summary of the thirty-three variables representing either a changed or maintained category, stratified by sex, is displayed in Table 1. Most respondents had not changed their behavior and BMI category at the time of the follow-up survey, with smoking and alcohol consumption having the smallest number of respondents who changed category. Physical activity

**Table 1. Categories of change or maintenance in BMI and health behaviors between baseline and follow-up surveys in the cohort sample and stratified by sex.**

| | Baseline | Follow-up | Total n | Total (%) | Men n | Men (%) | Women n | Women (%) |
|---|---|---|---|---|---|---|---|---|
| **Daily smoking** | Now | Now | 895 | 10.0 | 379 | 9.2 | 516 | 10.8 |
| | Now | Before | 661 | 7.4 | 296 | 7.2 | 365 | 7.6 |
| | Now | Never | 14 | 0.2 | 5 | 0.1 | 9 | 0.2 |
| | Before | Now | 152 | 1.7 | 76 | 1.8 | 76 | 1.6 |
| | Before | Before | 3 370 | 37.8 | 1 690 | 40.9 | 1 680 | 35.2 |
| | Before | Never | 254 | 2.9 | 118 | 2.9 | 136 | 2.8 |
| | Never | Now | 10 | 0.1 | 6 | 0.1 | 4 | 0.1 |
| | Never | Before | 125 | 1.4 | 53 | 1.3 | 72 | 1.5 |
| | Never | Never | 3 241 | 36.4 | 1 434 | 34.7 | 1 807 | 37.8 |
| | Missing | | 184 | 2.1 | 73 | 1.8 | 111 | 2.3 |
| **Alcohol consumption**[a] | High | High | 319 | 3.6 | 54 | 1.3 | 265 | 5.5 |
| | High | Low | 187 | 2.1 | 51 | 1.2 | 136 | 2.8 |
| | Low | High | 348 | 3.9 | 80 | 1.9 | 268 | 5.6 |
| | Low | Low | 7 693 | 86.4 | 3 817 | 92.4 | 3 876 | 81.2 |
| | Missing | | 359 | 4.0 | 128 | 3.1 | 231 | 4.8 |
| **Physical activity (min/week)** | ≥150 | ≥150 | 1 432 | 16.1 | 622 | 15.1 | 810 | 17.0 |
| | ≥150 | <150 | 851 | 9.6 | 339 | 8.2 | 512 | 10.7 |
| | <150 | ≥150 | 1 412 | 15.9 | 664 | 16.1 | 748 | 15.7 |
| | <150 | <150 | 4 352 | 48.9 | 2 144 | 51.9 | 2 208 | 46.2 |
| | Missing | | 859 | 9.6 | 361 | 8.7 | 498 | 10.4 |
| **BMI**[b] | Obese | Obese | 1 458 | 16.4 | 691 | 16.7 | 767 | 16.1 |
| | Obese | Overweight | 274 | 3.1 | 144 | 3.5 | 130 | 2.7 |
| | Obese | Normal | 9 | 0.1 | 4 | 0.1 | 5 | 0.1 |
| | Obese | Underweight | 0 | 0.0 | 0 | 0.0 | 0 | 0.0 |
| | Overweight | Obese | 612 | 6.9 | 282 | 6.8 | 330 | 6.9 |
| | Overweight | Overweight | 3 002 | 33.7 | 1 684 | 40.8 | 1 318 | 27.6 |
| | Overweight | Normal | 377 | 4.2 | 189 | 4.6 | 188 | 3.9 |
| | Overweight | Underweight | 1 | 0.0 | 0 | 0.0 | 1 | 0.0 |
| | Normal | Obese | 7 | 0.1 | 0 | 0.0 | 7 | 0.1 |
| | Normal | Overweight | 717 | 8.1 | 285 | 6.9 | 432 | 9.0 |
| | Normal | Normal | 2 345 | 26.3 | 834 | 20.2 | 1 511 | 31.6 |
| | Normal | Underweight | 32 | 0.4 | 2 | 0.0 | 30 | 0.6 |
| | Underweight | Obese | 0 | 0.0 | 0 | 0.0 | 0 | 0.0 |
| | Underweight | Overweight | 0 | 0.0 | 0 | 0.0 | 0 | 0.0 |
| | Underweight | Normal | 16 | 0.2 | 2 | 0.0 | 14 | 0.3 |
| | Underweight | Underweight | 23 | 0.3 | 3 | 0.1 | 20 | 0.4 |
| | Missing | | 33 | 0.4 | 10 | 0.2 | 23 | 0.5 |

[a] High alcohol consumption: More than 14 units per week for men and more than 7 units per week for women.

[b] Classification of weight status by body mass index (BMI): underweight (under 18.5 kg/m$^2$), normal weight (18.5 to under 25 kg/m$^2$), overweight (25 to under 30 kg/m$^2$) and obese (30 kg/m$^2$ and over).

and BMI had a larger number of respondents whose category changed at the time of the follow-up survey. The stratification by sex showed small relative differences among the portion of men and women who underwent changes in smoking, BMI, and physical activity. Regarding alcohol consumption, the percentage of women who changed their behavior was larger

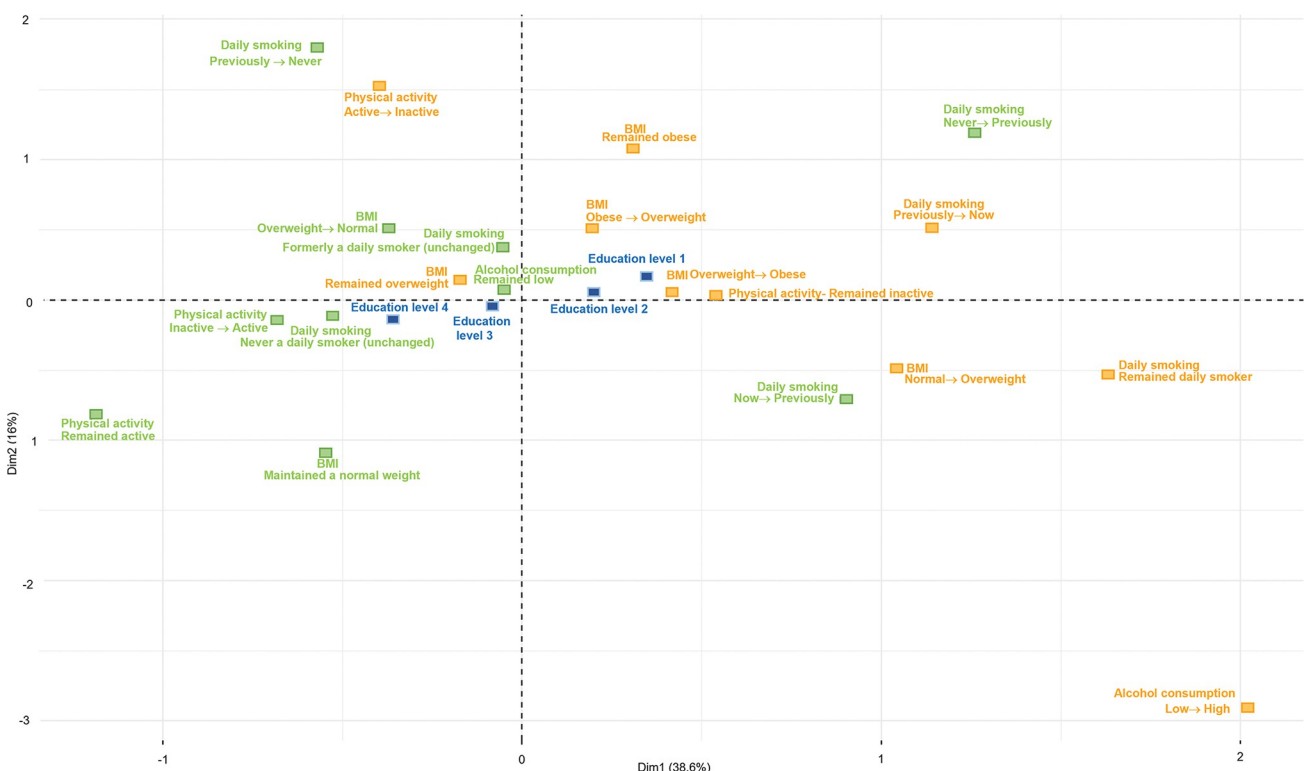

**Fig 1. MCA of BMI and health behavior patterns among men aged 32–47, with education as a supplementary variable.** Two-dimension plot of multiple correspondence among men aged 32–47 at baseline, 2007–2016. BMI: body mass index, normal weight: 18.5 to under 25 kg/m$^2$, overweight: 25 to under 30 kg/m$^2$, obese: 30 kg/m$^2$ and over. Alcohol consumption: high: more than 14 units per week, low: up to 14 units per week. Physical activity: active: 150 min/week or more, inactive: less than 150 min/week. Education: level 1: primary/partly secondary education (up to 10 years of schooling), level 2: upper secondary education (minimum of 3 years), level 3: college/university (less than 4 years), level 4: college/university (4 years or more).

compared to men, which can be partially explained by the higher threshold set for men to fall into the category of high alcohol consumption (fourteen or more units per week).

Figs 1 and 2 display the MCA plots for men, and Figs 3 and 4 presents the MCA plots for women. In the MCA, the axes or dimensions are interpreted by way of the contribution that each health behavior category makes to the total inertia, which is the term that describes the percentage of variability accounted for by the axis or dimension. The categories that contribute the most to the dimensions are the most significant in explaining the data set's variability, whereas the categories that are far from the origin indicate major differences between these combinations and the average. In the MCA of men, the inertia of the first two dimensions was 54.6% for the younger group (32–47 years of age at baseline); the first dimension explained 38.6% of data variability (visualized by the x-axis) and the second, 16% (y-axis). For the older group (48–87 years old at baseline), the inertia of the first two dimensions was 47.4%; the first dimension explained 28.5% of data variability and the second, 18.9%. In the MCA of women, the inertia of the first two dimensions was 51.5% for the younger group; the first dimension explained 36% of data variability (visualized by the x-axis) and the second, 15.5% (y-axis). For the older group, the inertia of the first two dimensions was 51.3%; the first dimension explained 35.8% of data variability and the second, 15.5%.

In all the MCA figures, the healthy (green) and unhealthy (orange) categories are positioned on opposite sides of the map, showing a clear distinction between the groups with higher education levels being associated with healthier categories and the unhealthier categories being associated with the groups with lower education levels. The MCA's visual output shows

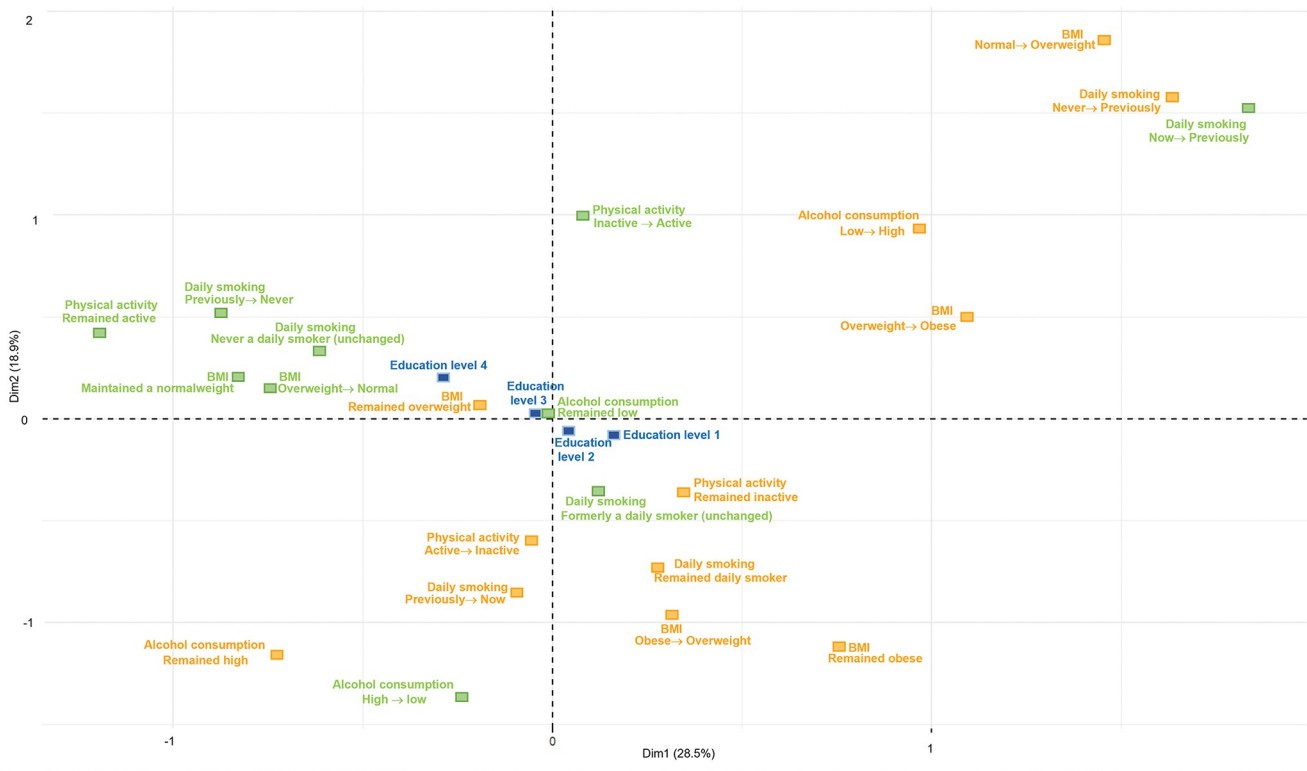

**Fig 2. MCA of BMI and health behavior patterns among men aged 48–87, with education as a supplementary variable.** BMI: body mass index, normal weight: 18.5 to under 25 kg/m$^2$, overweight: 25 to under 30 kg/m$^2$, obese: 30 kg/m$^2$ and over. Alcohol consumption: high: more than 14 units per week, low: up to 14 units per week. Physical activity: active: 150 min/week or more, inactive: less than 150 min/week. Education: level 1: primary/partly secondary education (up to 10 years of schooling), level 2: upper secondary education (minimum of 3 years), level 3: college/university (less than 4 years), level 4: college/university (4 years or more).

minimal, yet relevant differences between the age groups in both men and women. Among women, the differences between the first three education levels are smaller in the younger group. In this same group, a clear distinction can be seen between the patterns associated with the first three education levels and those associated with the highest education level group. The first three education levels are positioned on the left side of the map, indicating their association with a larger number of unhealthy patterns. The group with the highest education level appears separately on the opposite side of the map with a larger number of healthy categories, indicating that the highest education level is associated with healthier patterns. On the other hand, in the older group of women, there is a clear difference between the two lowest education levels and the other two groups with higher education levels. The two lowest education levels are associated with a larger number of unhealthy categories, whereas the higher education levels are associated with a larger number of healthy patterns. The opposite was observed among men, where the difference between the two lowest levels and the two highest levels was observed in the younger group, and the clustering of the first three levels was observed in the older group.

## Discussion

This study examined patterns of BMI, smoking, physical activity and alcohol consumption and investigated their association with education level, from 2008 to 2016 using longitudinal

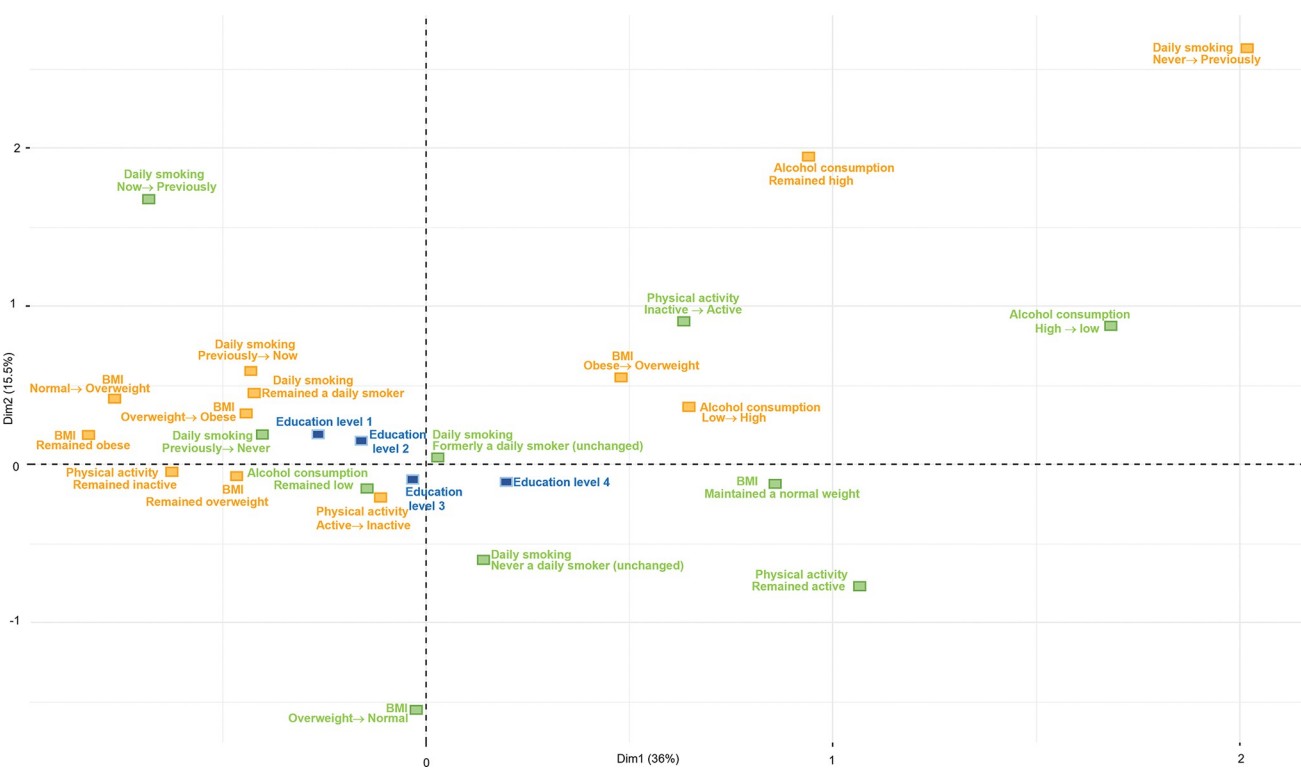

**Fig 3. MCA of BMI and health behavior patterns among women aged 32–47, with education as a supplementary variable.** Two-dimension plot of multiple correspondence among women aged 32–47 at baseline, 2007–2016. BMI: body mass index, normal weight: 18.5 to under 25 kg/m², overweight: 25 to under 30 kg/m², obese: 30 kg/m² and over. Alcohol consumption: high: more than 7 units per week, low: up to 7 units per week. Physical activity: active: 150 min/week or more, inactive: less than 150 min/week. Education: level 1: primary/partly secondary education (up to 10 years of schooling), level 2: upper secondary education (minimum of 3 years), level 3: college/university (less than 4 years), level 4: college/university (4 years or more).

data from a health survey in Norway. Most of the respondents did not change category of BMI and the three health behaviors between the baseline and follow-up surveys. Additionally, an educational gradient was found in these patterns, in which healthy changes and maintained healthy categories were associated with the highest educational levels. The main exception was high alcohol consumption, which was associated with higher education. With the exception of high alcohol consumption, our results were in line with a longitudinal study that followed multiple health behaviors among British men [20]. Moreover, they were similar to those reported in a Danish cohort study on several behaviors and risk factors such as obesity, in which those with high education levels had the highest alcohol intake levels [41]. A higher alcohol consumption has also been reported among groups with higher education levels in previous studies [42].

The results suggest individual's tendency to maintain their health behavior and BMI category as they transition through middle age. This tendency has also been observed in other studies in regard to smoking, physical activity and alcohol consumption [20], as well as in obesity [26]. In our study, while most participants maintained their behavior and BMI category between the two time points, the graphical representation of the MCA displayed a clear distinction between those with lower education levels and those with higher education levels in terms of healthy changes and maintenance of healthy categories. It appears that the groups with lower education are not only facing a higher prevalence of many unhealthy categories, but once they are exposed to both detrimental categories of BMI and health behavior, they remained exposed to them over a longer period.

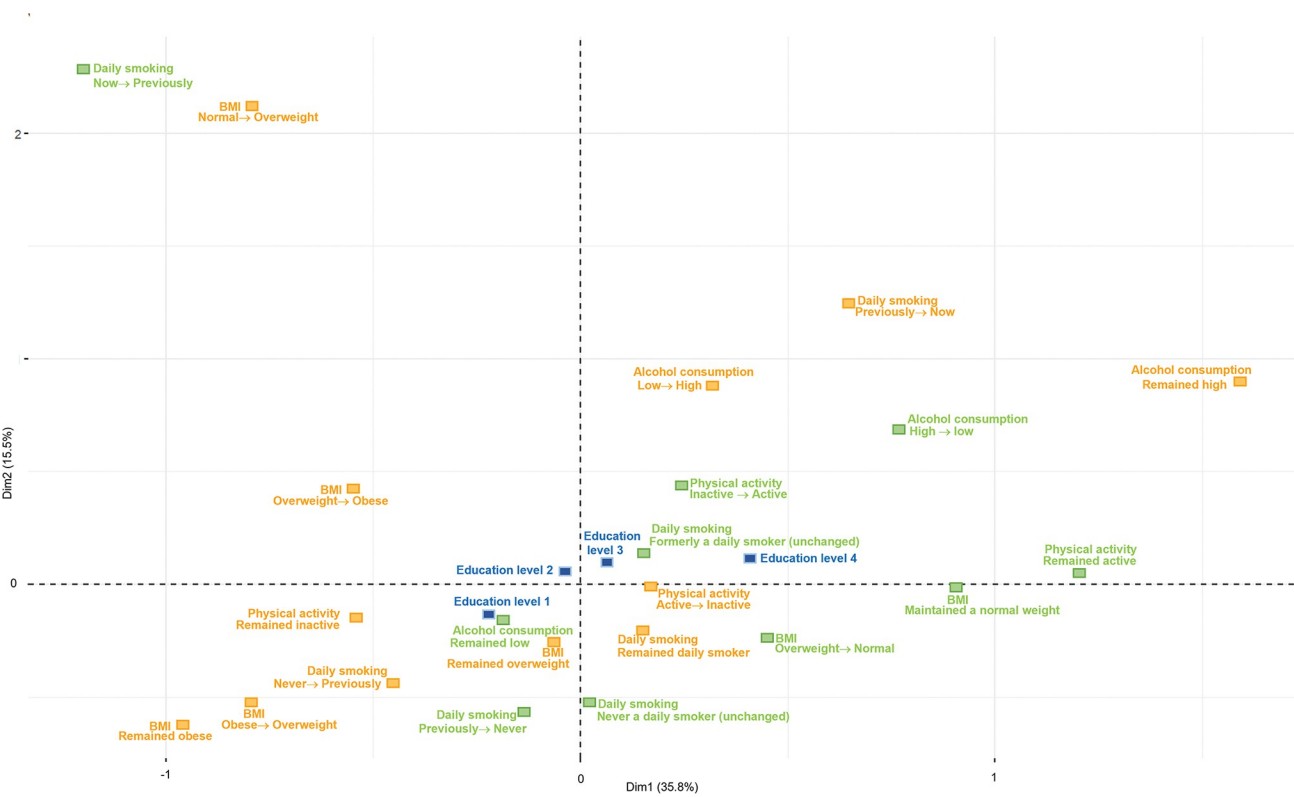

**Fig 4. MCA of BMI and health behavior patterns among women aged 48–87, with education as a supplementary variable.** BMI: body mass index, normal weight: 18.5 to under 25 kg/m², overweight: 25 to under 30 kg/m², obese: 30 kg/m² and over. Alcohol consumption: high: more than 7 units per week, low: up to 7 units per week. Physical activity: active: 150 min/week or more, inactive: less than 150 min/week. Education: level 1: primary/partly secondary education (up to 10 years of schooling), level 2: upper secondary education (minimum of 3 years), level 3: college/university (less than 4 years), level 4: college/university (4 years or more).

There is extensive literature about plausible mechanisms behind the well-known and complex relationships between education and health behaviors, and between education and BMI [43–46]. For example, according to the mechanism of differential exposure, an individual's socioeconomic position influences exposure to specific patterns, amounts, and duration of health risks [47]. Nevertheless, since follow-up studies on multiple trends of health behavior and BMI are rare, consistency has been hard to demonstrate. Another example is the mechanism of differential effects (also referred as differential vulnerability or susceptibility), which explains how the consequences of exposure to risk factors are also unevenly distributed across socioeconomic groups [45]. While the differential effects of exposure to risk factors across socioeconomic groups have been partly explained by interactions with other risk factors, the differences in effects have been observed even when all socioeconomic groups faced the same level of exposure [47, 48]. Findings from our follow-up study suggest that possibly, in addition to possible interactions with other unhealthy behavior factors—particularly among participants with lower education—a longer exposure time might be playing a significant role. Thus, socioeconomic differences in time of exposure to harmful combinations of health behaviors may also explain the differential effects across socioeconomic groups.

In Norway, possible country-specific explanations to the educational gradients in BMI and diverse health behaviors remain relatively unclear. For instance, a study that sought to examine whether educational differences in beliefs regarding the harms of smoking could explain the persistent educational gradient in smoking [49], the findings revealed no significant disparities

in these beliefs between individuals with lower and higher levels of education. This suggests that other factors are likely to play a role in the persistent and substantial educational disparities in tobacco smoking in Norway. Regarding BMI, a study about obesity and their association with level of education found that obesity was most common among low educated individuals [50]. The authors discussed the suitability of the diffusion theory of innovations [51] to describe the observed trends and how the ability to cope with low incentives to everyday physical activity and with the negative effects from environments where unlimited quantities of cheap high-energy food are available, might be highest among individuals with higher levels of education. In terms of physical activity, it has been found that physical activity taking place in natural environments is not only the most popular form of weekly physical activity, but also has been found to be related to higher levels of education [52].

On the other hand, the association between higher education and higher alcohol consumption may have different explanations in the Norwegian context. For example, the transition towards a Southern European drinking pattern occurring primarily among the higher educated in the population has been discussed to be a contributing factor [53].

Potential limitations of our analyses include selection bias, both in the Tromsø 6 participation alone and among those who participated in both the sixth and seventh waves of the Tromsø Study. For instance, 20.0% of the Tromsø 6 participants reported having more than four years of university education, while 22.4% of the respondents who participated in both waves reported the same. The increased proportion of respondents with higher education levels is a clear indication of a selection bias among those with the highest education level, adding to the selection bias previously shown for participation in Tromsø 6 [54]. The analyses excluded participants with missing data for BMI and the behavior variables, which might suggest selection bias due to the relationship between lower socioeconomic conditions and underreporting in health surveys [55]. Furthermore, the Tromsø Study is limited in terms of ethnic and minority diversity. While the largest proportion of indigenous populations live in Northern Norway, where the municipality of Tromsø is also located, more than 90% of the participants in the sixth wave of the Tromsø Study identified themselves as non-indigenous [54]. Among the remaining percentage, the large majority considered themselves as part of another ethnic group. The potential underrepresentation of the different ethnic groups in the study sample can also contribute to selection bias. In this regard, selection biases can lead to internally valid observations that cannot be generalized to the target population [56].

Another limitation is that almost all elements of the Tromsø study that are used in this study are self-reported, except for BMI, which was measured objectively at the time of each survey. However, education in the latest waves of the Tromsø Study has been recently validated by Vo and colleagues [57]. In addition to the potential bias introduced by self-reported information, the variables of physical activity and alcohol consumption were coded to align with current health guidelines. This process, which involved quantifying the responses to enable translation into "units per week" of alcohol consumption and "minutes per week" of physical activity has yet to be validated, and therefore can also contribute to measurement bias.

Moreover, our physical activity indicator does not provide information on intensity as recommended in current health guidelines [30, 32, 58]. Similarly, smoking behavior was limited to a single question inquiring about respondents' daily smoking habits. Although this approach allows for differentiation between daily and non-daily smokers, it does not account for volume of consumption or frequency of smoking beyond daily occurrences. Nonetheless, current health guidelines do not establish a safe threshold for smoking [30].

Furthermore, almost 3% of the respondents reported never having smoked daily in the follow-up survey, while they had previously reported smoking daily in the baseline survey. The respondents in this category were not removed from the analysis, as they may reflect another

group comprised of individuals who smoked daily on an occasional basis and did not perceive themselves as daily smokers, such as those who smoked only during social events [59]. Moreover, diet was excluded since dietary intake assessment through health surveys has major limitations [60].

Furthermore, despite the notable strengths of our study design, including its longitudinal design with a balanced panel and the establishment of educational attainment prior to the baseline survey, it is crucial to recognize that there may exist additional factors that could influence our findings. While education as a time-invariant variable enables the examination of trends in BMI and the health behaviors without the need to control for fluctuations in our measure of socioeconomic position, we have not fully accounted for other potentially influential factors. Specifically, factors such as income disparities [61] and variations in individuals' health status [62] have been demonstrated to exert an impact on health behavior factors. Nonetheless, income disparities in Norway are relatively minimal compared to other countries, which may mitigate the impact of salary on individuals' adherence to health recommendations [63]. In addition, it is important to also consider the reciprocal relationship between health behavior and income. In other words, while evidence highlights how income may shape health behavior factors, there is also evidence suggesting that health behavior factors can lead to income increases [64, 65]. Therefore, not only the influence of additional unmeasured variables must be considered, but also the direction of these relationships.

In conclusion, these findings highlight the extent and consistency of educational inequalities in the adherence to BMI categories and to multiple health behaviors related to health recommendations. This uneven distribution of both healthy changes and healthy categories that were maintained over time may drive the exacerbation of social inequalities in health and life expectancy. Our study also helped to shed light on the behaviors and BMI categories that are less prone to change among low educated individuals and can therefore be targeted by health interventions.

## Supporting information

**S1 Table. Characteristics of participants in Tromsø 6 and Tromsø 7 and the cohort sample.** [a] Percentage of participants in Tromsø 6 that also participated in Tromsø 7. [b] Percentage of participants in Tromsø 7 that also participated in Tromsø 6. High alcohol consumption: more than 14 units per week for men and 7 units per week for women. Low physical activity: Less than 150 minutes per week. Obesity: body mass index of 30 kg/m$^2$ or more.
(TIF)

## Acknowledgments

We are very grateful to Professor Michael Greenacre for providing expert opinion on the MCA.

## Author Contributions

**Conceptualization:** Ana Silvia Ibarra-Sanchez, Torbjørn Wisløff.

**Data curation:** Ana Silvia Ibarra-Sanchez.

**Formal analysis:** Ana Silvia Ibarra-Sanchez.

**Methodology:** Ana Silvia Ibarra-Sanchez, Torbjørn Wisløff.

**Software:** Ana Silvia Ibarra-Sanchez.

**Supervision:** Birgit Abelsen, Gang Chen, Torbjørn Wisløff.

**Validation:** Birgit Abelsen, Gang Chen, Torbjørn Wisløff.

**Visualization:** Ana Silvia Ibarra-Sanchez.

**Writing – original draft:** Ana Silvia Ibarra-Sanchez.

**Writing – review & editing:** Ana Silvia Ibarra-Sanchez, Birgit Abelsen, Gang Chen, Torbjørn Wisløff.

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
