## [Decision Letter · Decision Letter 0]

12 Oct 2022

PONE-D-22-22752Patterns of health behavior and socioeconomic status: a longitudinal multiple correspondence analysis of a middle-aged general population, 2007-2016PLOS ONE

Dear Dr. Ibarra-Sanchez,

Thank you for submitting your manuscript to PLOS ONE. After careful consideration, we feel that it has merit but does not fully meet PLOS ONE’s publication criteria as it currently stands. Therefore, we invite you to submit a revised version of the manuscript that addresses the points raised during the review process.

We look forward to receiving your revised manuscript.

Kind regards,

Diego Augusto Santos Silva, Ph.D.

Academic Editor

PLOS ONE

Journal Requirements:

Reviewers' comments:

Reviewer's Responses to Questions

**Comments to the Author**

1. Is the manuscript technically sound, and do the data support the conclusions?

Reviewer #1: Yes

Reviewer #2: Yes

Reviewer #3: Yes

2. Has the statistical analysis been performed appropriately and rigorously? 

Reviewer #1: Yes

Reviewer #2: Yes

Reviewer #3: No

3. Have the authors made all data underlying the findings in their manuscript fully available?

Reviewer #1: No

Reviewer #2: Yes

Reviewer #3: No

4. Is the manuscript presented in an intelligible fashion and written in standard English?

Reviewer #1: Yes

Reviewer #2: Yes

Reviewer #3: Yes

5. Review Comments to the Author

Reviewer #1: Overall comments

The entire manuscript needs proofreading—I noticed several spelling/grammar mistakes, and the writing could be improved. Certain sections need to be expanded (e.g. interpretation of results, exploration of mechanisms), while some could be shortened (e.g. how MCA works).

The figures/tables need to be revised, they are poorly organized, which makes them hard to follow For example, why was Table 1 stratified by gender, when gender was not discussed at all in the manuscript? And why was Table 2 never referenced? Is column percentage really the best way to present data?

It is unclear (at least based on the manuscript) if the changes in behaviors are measured at individual level or group level. And additional analyses might

Introduction

“Socioeconomic differences in health persist and are growing markedly, even in countries with good social circumstances (1-3).”

Please explain what you meant by social circumstances

“Unhealthy behaviors, such as smoking, harmful alcohol consumption, physical inactivity, poor diet and high BMI are risk factors for chronic diseases (7, 8)”

High BMI is not a behavior; and it is more like an intermediate or measure of outcome in your case

“Monitoring social inequalities in the burden of chronic diseases and its determinants can facilitate the development of policies to improve health equality”

Please proofread the entire manuscript

Materials and methods

“The Tromsø Study is a prospective cohort of the residents of the municipality of Tromsø”

Just to confirm, these are the same people followed over time? I think you mentioned it later but you need to clarify it here too. Is it an open cohort? What are the retention rate? Etc.

“Level of education was used as indicator of socioeconomic status due to previous research showing education as one of the main indicators of socioeconomic status in the Nordic countries (35)”

I would like to see more justification of using only education as indicator, since socioeconomic status is fairly complex and could be measured in many ways. You might want to compare several measures since it is the key exposure/factor in your study

Results

“Only 6.3% of the respondents reported a change in alcohol consumption behavior, followed by 13.1% in daily smoking.”

Please only describe things you meant to interpret—here you combined both health and unhealthy changes—which might be pointless to bring up

“In dimension 1, BMI and BMI change are indicated to have the largest discriminatory power, with “remain normal weight” on one side, and three different unhealthy BMI categories on the right side (“remain obese”, “overweight to obese” and “normal to overweight”).”

There was a very long section on MCA in the methods section, yet here there was no explanation on what the dimensions mean. Need to expand this section for those who aren’t familiar with MCA.

“Level of education was included as a supplementary variable. Supplementary points define additional profiles that are not used to establish the solution space but are projected onto the space afterwards.”

Does that mean this study is purely descriptive? You might want to consider additional analyses like regression where you could account for confounding. Also, since there’s no measure of association, you need to be cautious when discussing biases in later section.

Discussion

“A longer exposure time could further explain the mechanism of differential susceptibility (47)”

The mechanism was barely explored in the manuscript

Reviewer #2: This is an interesting manuscript with a novel approach to measuring the mediators of health inequalities through the life course.

The authors’ rationale is clear, as are the results here-presented. However, I believe the manuscript has some limitations that need to be revised and some gaps that need to be filled in.

Introduction:

1) When mentioning health differences, in line 56, please specify which differences (all-cause mortality) as the size of inequalities could vary according to the health indicator. Also, mention how these differences are measured: rate ratios vs rate differences.

2) In line 63 the authors state that little is known regarding inequalities in multiple health behaviours. The authors must consult Lakshman R, et al. article (Lakshman R, et al. Association between area-level socioeconomic deprivation and a cluster of behavioural risk factors: cross-sectional, population-based study. Journal of Public Health. 2010 Sep 29;33(2):234-45)

3) In lines 69 to 72 the authors seem to oversimplify (and incorrectly report) the reference:

[2003] “When compared to the Healthy group, individuals in the Apathetic group were younger (odds ratio [OR] = 0.92), male (OR = 2.89), lower income (OR = 0.87), less educated (OR = 0.74), more likely to be Black (OR = 1.30), and less likely to be Other (OR = 0.79) versus White. The Binge-drinking group was younger (OR = 0.72), male (OR = 7.38), lower income (OR = 0.85), less educated (OR = 0.57), and less likely to be Hispanic (OR = 0.76) versus White than the Healthy group.

[2015] “Participants in the Apathetic group were more likely to be younger (OR = 0.87), male (OR = 1.41), lower income (OR = 0.63), less educated (OR = 0.45), Black versus White (OR = 1.86), and White versus Hispanic (OR = 0.67) than the Physically Active group. Participants in the Binge-drinking group were younger (OR = 0.73), male (OR = 3.28), lower education (OR = 0.53), lower income (OR = 0.79), Black versus White (OR = 1.31), White versus Hispanic (OR = 0.70), and Other (OR = 0.66) when compared to the Physically Active group.”

As such these findings should be reported differently: (1) by behaviour group (healthy/physically active vs apathetic vs binge drinking, and mentioning what behaviours characterize these groups), and (2) reporting the results from the binge drinking group.

Further, the authors must confront these findings from the US to those from other authors and studies, as from Cutler et al. (Cutler DM, Lleras-Muney A. Understanding differences in health behaviors by education. Journal of health economics. 2010;29(1):1-28)

4) Considering lines 84-85 (“observations from longitudinal studies have suggested that a large percentage of individuals follow a pattern of long-term adherence to the same health behaviors”) the authors must clearly state (1) what they expect this study to add to the literature and (2) their research hypothesis.

Methods:

1) The use of BMI as a proxy for healthy eating seems inappropriate, as BMI importantly depends not only on the diet but also on the intensity and frequency of physical activity, among others. It seems more adequate to remove this variable from the study or assume it as BMI (and not a proxy).

2) It is unclear if the authors grouped the categories of the variables of interest or if they used the original categorization. The authors must explain the rationale behind the decision of aggregation/no aggregation of these categories (for example, is low BMI considered normal weight?).

3) The authors must state if (and how) sex and age were used in the multiple correspondence analysis.

Results:

1) The authors must clarify what means, for the study aim, (1) the inertia of the two dimensions, (2) BMI’s “largest discriminatory power”, (3) the contributions of physical activity and alcohol consumption to the spread of dimension 1, (4) the considerable discriminatory power of BMI and smoking on dimension 2.

2) A legend is needed in Figure 1.

Discussion:

1) Besides the ‘differential susceptibility’ other factors and pathways may explain (1) the higher risk of low-educated individuals to have unhealthy behaviours and (2) their difficulty in changing them, besides the (3) overall population to change behaviours. Material and immaterial resources (see Mackenbah et al. The persistence of health inequalities in modern welfare states: The explanation of a paradox. SSM 2012), psychosocial stress and available coping mechanisms, environmental opportunities, the capacity of the health system to support people and their knowledge on how to navigate it, social norms and social control, can be some of the hypotheses that should be enunciated in this section.

2) The reader probably will know little about the Norwegian context: it would be important that the authors contextualize some of the reasons behind the persistence of unhealthy behaviours in the life course, especially among the least educated.

3) The authors mention a selection bias regarding education - couldn’t there be a bias regarding behaviours, ie., couldn’t people with a higher number of unhealthy behaviours refrain from participating in subsequent waves?

4) How may these selection biases impact this study's results?

5) As stated above, BMI should not be used as an indicator of healthy eating. It strongly depends on physical activity (besides other factors).

Reviewer #3: This is a two wave survey (longitudinal) on 4-5 health behaviors and one SES indicator being educational attainment. While past predicts future, healthy behaviors tend to covary, and they are associated with higher educational attainment. These are partially known, but that does not make me less interested in the results if:

1- The paper discusses that overlap between SES and various health behaviors suggest there might be a sub-additive effects of various health behaviors, because of overlapping mechanism. That means, the total effect is probably smaller than sum of the effects, because health behaviors tend to manifest in the same individual. This subadditive versus multiplicative/synergistic effect if of interest of the literature.

2- The paper needs to go beyond main effects that assume all effects are universal across subgroups. We did not see the distribution of immigrants and native individuals and ethnic groups. A large body of literature shown that the effects of educationn is maximum in native people and minimum in marginalized people such as immigrants because the system does not similarly value their education, and their education does not become income etc whoch is needed for healty diet and exercise. so, these diminished returns of education based on marginalization status should be tested. If all associations are universal (no interaction), then your country is a very non-discriminatory context, but if marginalized people with high education still engage in health risk behaviors, it is probably because they work in worse jobs and have lower income nad higher stress. These to be tested and discussed based on a very well-established litertaure on MDRs (diminished returns). Nothing should be assumed to be universal. One size does not fit all.

After these comments are addressed, I can review the paper again and suggest publication.

6. PLOS authors have the option to publish the peer review history of their article (what does this mean?). If published, this will include your full peer review and any attached files.

Reviewer #1: No

Reviewer #2: No

Reviewer #3: No

---

## [Author Response · Author response to Decision Letter 0]

20 Dec 2022

Reviewer #1: Overall comments

1. The entire manuscript needs proofreading—I noticed several spelling/grammar mistakes, and the writing could be improved. Certain sections need to be expanded (e.g. interpretation of results, exploration of mechanisms), while some could be shortened (e.g. how MCA works).

Thanks for this comment. We have now revised the language and proofread the manuscript. We have expanded our interpretation of results, mechanisms and shortened the section on how Multiple Correspondence Analysis works. 

2. The figures/tables need to be revised, they are poorly organized, which makes them hard to follow For example, why was Table 1 stratified by gender, when gender was not discussed at all in the manuscript? And why was Table 2 never referenced? Is column percentage really the best way to present data?

Figures and tables have been revised according to the organization of the revised manuscript, assured they were referenced in the main body of the text. In addition, we have added an additional analysis that explore sex differences.

3. It is unclear (at least based on the manuscript) if the changes in behaviors are measured at individual level or group level. And additional analyses might

Thanks for making us aware of this! We have now elaborated on this in our methods section, lines [159-161] , making it clearer that changes are measured at the individual level. 

4. Introduction

“Socioeconomic differences in health persist and are growing markedly, even in countries with good social circumstances (1-3).”

Please explain what you meant by social circumstances.

We have reworded this sentence to explain what we meant by social circumstances. See line no. [45-46].

5. “Unhealthy behaviors, such as smoking, harmful alcohol consumption, physical inactivity, poor diet and high BMI are risk factors for chronic diseases (7, 8)”

High BMI is not a behavior; and it is more like an intermediate or measure of outcome in your case

Thanks for pointing this out. We have changed this sentence to make this distinction clearer throughout the manuscript. 

6. “Monitoring social inequalities in the burden of chronic diseases and its determinants can facilitate the development of policies to improve health equality”

Please proofread the entire manuscript.

Thanks for this comment. We have now revised the language and proofread the manuscript.

7. Materials and methods

“The Tromsø Study is a prospective cohort of the residents of the municipality of Tromsø”

Just to confirm, these are the same people followed over time? I think you mentioned it later but you need to clarify it here too. Is it an open cohort? What are the retention rate? Etc.

We are very grateful for this comment. More detail about the Tromsø study has been added to clarify that we are following the same people over time, and the retention rate can be seen in the table of descriptive characteristics. 

8.“Level of education was used as indicator of socioeconomic status due to previous research showing education as one of the main indicators of socioeconomic status in the Nordic countries (35)”

I would like to see more justification of using only education as indicator, since socioeconomic status is fairly complex and could be measured in many ways. You might want to compare several measures since it is the key exposure/factor in your study. 

Good points! In terms of education as an indicator of SEP, we have now added information to justify it as indicator of socioeconomic status, lines [136-144]. For example, education is a stable measure that is maintained even if respondents change their employment status over time, which is ideal for the design of our study. An additional advantage is that the education variable in the lates waves of the Tromsø Study has been recently validated. Since the variable of income remains self-reported to this date, we could not add a suitable comparison between several measures of SEP. 

9. Results

“Only 6.3% of the respondents reported a change in alcohol consumption behavior, followed by 13.1% in daily smoking.”

Please only describe things you meant to interpret—here you combined both health and unhealthy changes—which might be pointless to bring up

Good point! We have now distinguished between the behavior factors that underwent more change versus those that changed less. Interestingly, smoking and alcohol consumption were the questions in which a larger number of respondents in the cohort sample reported the same answer in 2007/08 and in 2015/16. In contrast, a larger number of respondents reported a different answer or fell into a different category with respect to BMI and physical activity at the follow-up. We consider this an important observation and therefore discuss this further on lines [176-184].

10. “In dimension 1, BMI and BMI change are indicated to have the largest discriminatory power, with “remain normal weight” on one side, and three different unhealthy BMI categories on the right side (“remain obese”, “overweight to obese” and “normal to overweight”).”

There was a very long section on MCA in the methods section, yet here there was no explanation on what the dimensions mean. Need to expand this section for those who aren’t familiar with MCA.

Thanks for making us aware of this. We have now reworded our results section by removing the overtechnical use of language used in the field of MCA and have now kept only the information most relevant to the main findings for our results’ interpretation.

11. “Level of education was included as a supplementary variable. Supplementary points define additional profiles that are not used to establish the solution space but are projected onto the space afterwards.”

Does that mean this study is purely descriptive? You might want to consider additional analyses like regression where you could account for confounding. Also, since there’s no measure of association, you need to be cautious when discussing biases in later section. 

Thank you for pointing out this about confounding! We could account now for some of the confounding in the complex relationship between education and health behavior by stratifying the analysis by age and gender. Although we don’t perform any specific hypothesis testing in the manuscript, our study is not deemed as purely descriptive, see line [152-157]. MCA is a visualization of the measure of associations between a set of variables. We will look further into other statistical ways of analyzing this relationship in future work, as also noted in the discussion.

12. Discussion

“A longer exposure time could further explain the mechanism of differential susceptibility (47)”

The mechanism was barely explored in the manuscript.

We have expanded our discussion section with regard to this concept. 

Reviewer #2: 

13.This is an interesting manuscript with a novel approach to measuring the mediators of health inequalities through the life course.

The authors’ rationale is clear, as are the results here-presented. However, I believe the manuscript has some limitations that need to be revised and some gaps that need to be filled in.

Introduction:

When mentioning health differences, in line 56, please specify which differences (all-cause mortality) as the size of inequalities could vary according to the health indicator. Also, mention how these differences are measured: rate ratios vs rate differences.

Thanks for pointing this out. We have now included more information about the European comparison by Mackenback and colleagues.

14. In line 63 the authors state that little is known regarding inequalities in multiple health behaviours. The authors must consult Lakshman R, et al. article (Lakshman R, et al. Association between area-level socioeconomic deprivation and a cluster of behavioural risk factors: cross-sectional, population-based study. Journal of Public Health. 2010 Sep 29;33(2):234-45)

Thanks for making us aware of this important work by Lakshman and colleagues! We agree that little is not the appropriate term, therefore in line 63 we originally wrote less is known. There is less research with longitudinal design about socioeconomic inequalities in multiple health behaviors compared to the number of cross-sectional studies. We could not fit this work by Lakshman and colleagues because we wanted to focus on highlighting the studies with longitudinal or repeated cross-sectional design and/or from Scandinavian populations. 

15. In lines 69 to 72 the authors seem to oversimplify (and incorrectly report) the reference:

[2003] “When compared to the Healthy group, individuals in the Apathetic group were younger (odds ratio [OR] = 0.92), male (OR = 2.89), lower income (OR = 0.87), less educated (OR = 0.74), more likely to be Black (OR = 1.30), and less likely to be Other (OR = 0.79) versus White. The Binge-drinking group was younger (OR = 0.72), male (OR = 7.38), lower income (OR = 0.85), less educated (OR = 0.57), and less likely to be Hispanic (OR = 0.76) versus White than the Healthy group.

[2015] “Participants in the Apathetic group were more likely to be younger (OR = 0.87), male (OR = 1.41), lower income (OR = 0.63), less educated (OR = 0.45), Black versus White (OR = 1.86), and White versus Hispanic (OR = 0.67) than the Physically Active group. Participants in the Binge-drinking group were younger (OR = 0.73), male (OR = 3.28), lower education (OR = 0.53), lower income (OR = 0.79), Black versus White (OR = 1.31), White versus Hispanic (OR = 0.70), and Other (OR = 0.66) when compared to the Physically Active group.”

Thank you for expanding this for us. We have now expanded the findings from this research article and corrected our reporting of these results. 

16. Further, the authors must confront these findings from the US to those from other authors and studies, as from Cutler et al. (Cutler DM, Lleras-Muney A. Understanding differences in health behaviors by education. Journal of health economics. 2010;29(1):1-28)

Thanks for making us aware of this interesting study by Cutler and Lleras-Muney. We have now discussed our findings related to this and other work. 

17. Considering lines 84-85 (“observations from longitudinal studies have suggested that a large percentage of individuals follow a pattern of long-term adherence to the same health behaviors”) the authors must clearly state (1) what they expect this study to add to the literature and (2) their research hypothesis.

We have now clearly stated what we expect our study will add to the current literature in line no. [63-67] and [86-92]. 

18. Methods:

The use of BMI as a proxy for healthy eating seems inappropriate, as BMI importantly depends not only on the diet but also on the intensity and frequency of physical activity, among others. It seems more adequate to remove this variable from the study or assume it as BMI (and not a proxy).

Thanks for this insight! We have now made sure BMI is not assumed as a proxy for healthy eating and now is assumed it as BMI.

19. It is unclear if the authors grouped the categories of the variables of interest or if they used the original categorization. The authors must explain the rationale behind the decision of aggregation/no aggregation of these categories (for example, is low BMI considered normal weight?).

Thanks for pointing out that there was some justification missing in our methods section. We have now added information about the way categories were grouped in the methods section. 

20. The authors must state if (and how) sex and age were used in the multiple correspondence analysis.

We have now stated how age and sex were used in the analyses.

21. Results:

1) The authors must clarify what means, for the study aim, (1) the inertia of the two dimensions, (2) BMI’s “largest discriminatory power”, (3) the contributions of physical activity and alcohol consumption to the spread of dimension 1, (4) the considerable discriminatory power of BMI and smoking on dimension 2.

We have now reworded our results section to remove the overtechnical use of language and keeping only the information most relevant to the main findings for our results’ interpretation.

22. A legend is needed in Figure 1.

We have revised our figures including their legends.

23. Discussion:

1) Besides the ‘differential susceptibility’ other factors and pathways may explain (1) the higher risk of low-educated individuals to have unhealthy behaviours and (2) their difficulty in changing them, besides the (3) overall population to change behaviours. Material and immaterial resources (see Mackenbah et al. The persistence of health inequalities in modern welfare states: The explanation of a paradox. SSM 2012), psychosocial stress and available coping mechanisms, environmental opportunities, the capacity of the health system to support people and their knowledge on how to navigate it, social norms and social control, can be some of the hypotheses that should be enunciated in this section.

Thanks for pointing us in the direction of this important work. We have now mentioned the literature behind the diverse pathways in our discussion section.

24. The reader probably will know little about the Norwegian context: it would be important that the authors contextualize some of the reasons behind the persistence of unhealthy behaviors in the life course, especially among the least educated.

Thanks for pointing us in the direction of this important work. Literature about the mechanisms behind the relationship between education and health behavior in the Norwegian context has been added to our discussion. 

25. The authors mention a selection bias regarding education - couldn’t there be a bias regarding behaviours, ie., couldn’t people with a higher number of unhealthy behaviours refrain from participating in subsequent waves?

Excellent point, we have mentioned this on lines [312-317].

26. How may these selection biases impact this study's results?

We have now included more about the potential impact of the selection bias.

27. As stated above, BMI should not be used as an indicator of healthy eating. It strongly depends on physical activity (besides other factors).

Thanks for this insight! We have now made sure BMI is not assumed as a proxy for healthy eating and now is assumed it as BMI.

Reviewer #3: 

28. This is a two wave survey (longitudinal) on 4-5 health behaviors and one SES indicator being educational attainment. While past predicts future, healthy behaviors tend to covary, and they are associated with higher educational attainment. These are partially known, but that does not make me less interested in the results if:

1- The paper discusses that overlap between SES and various health behaviors suggest there might be a sub-additive effects of various health behaviors, because of overlapping mechanism. That means, the total effect is probably smaller than sum of the effects, because health behaviors tend to manifest in the same individual. This subadditive versus multiplicative/synergistic effect if of interest of the literature.

Thanks for this important insight! We have now welcomed the literature about this point in our discussion section. 

29. 2- The paper needs to go beyond main effects that assume all effects are universal across subgroups. We did not see the distribution of immigrants and native individuals and ethnic groups. A large body of literature shown that the effects of education is maximum in native people and minimum in marginalized people such as immigrants because the system does not similarly value their education, and their education does not become income etc whoch is needed for healty diet and exercise. so, these diminished returns of education based on marginalization status should be tested. If all associations are universal (no interaction), then your country is a very non-discriminatory context, but if marginalized people with high education still engage in health risk behaviors, it is probably because they work in worse jobs and have lower income nad higher stress. These to be tested and discussed based on a very well-established litertaure on MDRs (diminished returns). Nothing should be assumed to be universal. One size does not fit all.

After these comments are addressed, I can review the paper again and suggest publication.

We completely agree that this is a very important issue to take into account. We have now mentioned this aspect in the discussion section .

---

## [Decision Letter · Decision Letter 1]

4 May 2023

PONE-D-22-22752R1Health behavior patterns and socioeconomic status: a longitudinal multiple correspondence analysis of a middle-aged general population, 2007-2016PLOS ONE

Dear Dr. Ibarra-Sanchez,

Thank you for submitting your manuscript to PLOS ONE. After careful consideration, we feel that it has merit but does not fully meet PLOS ONE’s publication criteria as it currently stands. Therefore, we invite you to submit a revised version of the manuscript that addresses the points raised during the review process.

We look forward to receiving your revised manuscript.

Kind regards,

Diego Augusto Santos Silva, Ph.D.

Academic Editor

PLOS ONE

Additional Editor Comments :

Based on the reviewer's comment, authors should note the following points in the article:

Strengths

• This was an interesting study that longitudinally assessed multiple health behaviors in the same study sample over time that included both men and women.

• The longitudinal design, large sample size, and decision to assess multiple health behaviours were strengths.

• MCA analysis was novel.

Weaknesses

• Introduction page 12, line 48: high BMI is not a behaviour as the others are, but is a clinical risk factor that is the result of poor diet and low physical activity - if you mention BMI here you should also mention high blood pressure, cholesterol etc. Same for line 56. If the focus of the paper is to highlight socioeconomic differences in health behaviours, BMI should not be included in this list.

• Related to above, the authors continue to refer to BMI as a poor health behaviour, which it is not. It is the result of poor health behaviours (poor diet and low levels of physical activity) – and in some cases, BMI does not reflect either of these (e.g., in athletes with high muscle mass relative to height). Further, BMI (weight) may increase over time not due to poor health behaviours, but medication side effects and hormonal changes associated with pregnancy or menopause (in women). This is another reason why it is conceptually and methodologically inappropriate to define BMI as a health behaviour, so any reference to BMI being a health behaviour should be deleted and/or edited throughout the manuscript. The previous reviewers had also made this comment but this has not been addressed by the authors.

• The main outcome measures (health behaviours) were not very sensitive. For example, smoking was assessed as current, past or never – but did not include a measure of volume or duration, which is critical for estimating impacts on health. Pack years would have been a more sensitive measure. Also, participants were defined as physically active only if they reached the 150min/week threshold, but this obscures the ability to observe a dose-response relationship between education and physical activity. Further, the authors calculated physical activity minutes based on multiplying reported frequency and duration, but they estimated duration based on categorical data – this is not appropriate unless there is validation information for this?

• The decision to measure SES by education alone was confusing. The authors reasoned that this would be a good measure because it is stable, yet their study was longitudinal, so if their predictor is stable, what is the benefit of a longitudinal analyses, as this would make it less likely to observe changes in health behaviours as a result of changes in SES over time (because education does not change like occupation and income, which could affect the ability to ‘afford’ engaging in good health behaviours). The authors provided no hypotheses - is this what they expected to observe? This is indeed what they found, but this seems entirely predictable based on previous work and the nature/stability of this variable. What would have been more interesting is to examine interactions between education (stable variable) and occupation or income (unstable variable) which might help tease apart how these variables contribute to health behaviours. For example, education might be important for healthy eating, except when one loses their job and income – where buying healthy foods might be more difficult. Similarly, education might be associated with higher alcohol consumption (because drinking can be expensive and it is often seen as status symbol)- yet this association might decrease in those who lose their job or income.

• Table 1 was confusing and hard to interpret because of the high number of independent groups that were presented across each health behaviour (and BMI is not a health behaviour). It would be more helpful and informative to classify each behaviour as getting better, worse or staying the same over time as a function of sex.

• I was not able to read the figures because the resolution was not high enough. The text summarizing the results of the MCA analysis was also confusing – possibly because of the multiple analyses and largely descriptive nature of the findings.

• The discussion did not explain any of the mechanisms behind the observed associations, which really diminishes the contribution of the manuscript. The text on page 21, line 281 to 305, seems to go in circles and say no more than education is tied to health behaviours and this is generally maintained over time. How this is association may be explained by interactions with other risk factors was not discussed.

• As the authors acknowledged, there is a high risk of selection bias and self-report bias, which undermines confidence in the data.

Reviewers' comments:

Reviewer's Responses to Questions

**Comments to the Author**

1. If the authors have adequately addressed your comments raised in a previous round of review and you feel that this manuscript is now acceptable for publication, you may indicate that here to bypass the “Comments to the Author” section, enter your conflict of interest statement in the “Confidential to Editor” section, and submit your "Accept" recommendation.

Reviewer #1: All comments have been addressed

Reviewer #4: (No Response)

Reviewer #5: All comments have been addressed

2. Is the manuscript technically sound, and do the data support the conclusions?

Reviewer #1: Yes

Reviewer #4: Partly

Reviewer #5: Yes

3. Has the statistical analysis been performed appropriately and rigorously? 

Reviewer #1: Yes

Reviewer #4: I Don't Know

Reviewer #5: Yes

4. Have the authors made all data underlying the findings in their manuscript fully available?

Reviewer #1: Yes

Reviewer #4: Yes

Reviewer #5: Yes

5. Is the manuscript presented in an intelligible fashion and written in standard English?

Reviewer #1: Yes

Reviewer #4: Yes

Reviewer #5: Yes

6. Review Comments to the Author

Reviewer #1: The authors have addressed my comments--justifications were provided for each step of the analysis, and tables/writing have been revised. As a result, the manuscript improved noticeably.

Reviewer #4: Summary

This is a resubmission of an article assessing patterns of multiple health behaviour factor and their association with SES (defined as education level) in two waves of a longitudinal cohort (2007-08 and 2015-16) of 8906 adults living in Northern Norway. Main findings were that healthier behaviour patterns were observed in those with more education. Also, positive changes in health behaviours and maintenance of good health behaviours was associated with higher education levels. The authors concluded that policy makers should examine different opportunities for engaging in good health behaviours as a function of SES, which should be addressed to tackle health inequalities.

I did not review the original submission, but carefully reviewed the responses to reviews and edits made the manuscript. It would seem that some issues have been clarified or addressed, but other were either not addressed or not addressed appropriately based on my interpretation of the reviews. I have detailed my assessment the manuscripts main strengths and weaknesses, and where further edits and clarifications are needed. Overall, based on my reading of the manuscript, I am not sure it makes a strong or unique enough contribution to the extant literature to warrant publication in PLOS One. I recognized that the authors have done an extensive response to review and made several edits to the manuscript, but the results are largely descriptive and there remain important conceptual and methodological issues with the paper. I have summarized the paper’s major strengths and weaknesses below.

Strengths

• This was an interesting study that longitudinally assessed multiple health behaviors in the same study sample over time that included both men and women.

• The longitudinal design, large sample size, and decision to assess multiple health behaviours were strengths.

• MCA analysis was novel.

Weaknesses

• Introduction page 12, line 48: high BMI is not a behaviour as the others are, but is a clinical risk factor that is the result of poor diet and low physical activity - if you mention BMI here you should also mention high blood pressure, cholesterol etc. Same for line 56. If the focus of the paper is to highlight socioeconomic differences in health behaviours, BMI should not be included in this list.

• Related to above, the authors continue to refer to BMI as a poor health behaviour, which it is not. It is the result of poor health behaviours (poor diet and low levels of physical activity) – and in some cases, BMI does not reflect either of these (e.g., in athletes with high muscle mass relative to height). Further, BMI (weight) may increase over time not due to poor health behaviours, but medication side effects and hormonal changes associated with pregnancy or menopause (in women). This is another reason why it is conceptually and methodologically inappropriate to define BMI as a health behaviour, so any reference to BMI being a health behaviour should be deleted and/or edited throughout the manuscript. The previous reviewers had also made this comment but this has not been addressed by the authors.

• The main outcome measures (health behaviours) were not very sensitive. For example, smoking was assessed as current, past or never – but did not include a measure of volume or duration, which is critical for estimating impacts on health. Pack years would have been a more sensitive measure. Also, participants were defined as physically active only if they reached the 150min/week threshold, but this obscures the ability to observe a dose-response relationship between education and physical activity. Further, the authors calculated physical activity minutes based on multiplying reported frequency and duration, but they estimated duration based on categorical data – this is not appropriate unless there is validation information for this?

• The decision to measure SES by education alone was confusing. The authors reasoned that this would be a good measure because it is stable, yet their study was longitudinal, so if their predictor is stable, what is the benefit of a longitudinal analyses, as this would make it less likely to observe changes in health behaviours as a result of changes in SES over time (because education does not change like occupation and income, which could affect the ability to ‘afford’ engaging in good health behaviours). The authors provided no hypotheses - is this what they expected to observe? This is indeed what they found, but this seems entirely predictable based on previous work and the nature/stability of this variable. What would have been more interesting is to examine interactions between education (stable variable) and occupation or income (unstable variable) which might help tease apart how these variables contribute to health behaviours. For example, education might be important for healthy eating, except when one loses their job and income – where buying healthy foods might be more difficult. Similarly, education might be associated with higher alcohol consumption (because drinking can be expensive and it is often seen as status symbol)- yet this association might decrease in those who lose their job or income.

• Table 1 was confusing and hard to interpret because of the high number of independent groups that were presented across each health behaviour (and BMI is not a health behaviour). It would be more helpful and informative to classify each behaviour as getting better, worse or staying the same over time as a function of sex.

• I was not able to read the figures because the resolution was not high enough. The text summarizing the results of the MCA analysis was also confusing – possibly because of the multiple analyses and largely descriptive nature of the findings.

• The discussion did not explain any of the mechanisms behind the observed associations, which really diminishes the contribution of the manuscript. The text on page 21, line 281 to 305, seems to go in circles and say no more than education is tied to health behaviours and this is generally maintained over time. How this is association may be explained by interactions with other risk factors was not discussed.

• As the authors acknowledged, there is a high risk of selection bias and self-report bias, which undermines confidence in the data.

Reviewer #5: The authors seemed to have made reasonable efforts in addressing all of the reviewers' comments they have received in their previous iteration of submission.

7. PLOS authors have the option to publish the peer review history of their article (what does this mean?). If published, this will include your full peer review and any attached files.

Reviewer #1: No

Reviewer #4: **Yes: **Kim L. Lavoie

Reviewer #5: No

---

## [Author Response · Author response to Decision Letter 1]

28 Jun 2023

Strengths

• This was an interesting study that longitudinally assessed multiple health behaviors in the same study sample over time that included both men and women.

• The longitudinal design, large sample size, and decision to assess multiple health behaviours were strengths.

• MCA analysis was novel.

Thank you for acknowledging the strengths of our research study.

Weaknesses

• Introduction page 12, line 48: high BMI is not a behaviour as the others are, but is a clinical risk factor that is the result of poor diet and low physical activity - if you mention BMI here you should also mention high blood pressure, cholesterol etc. Same for line 56. If the focus of the paper is to highlight socioeconomic differences in health behaviours, BMI should not be included in this list.

Thank you for your comment. As you point out, it is important that BMI is not referred as a behavior. We have now made the necessary changes throughout the paper to make it clearer that BMI is not referred as a behavior.

• Related to above, the authors continue to refer to BMI as a poor health behaviour, which it is not. It is the result of poor health behaviours (poor diet and low levels of physical activity) – and in some cases, BMI does not reflect either of these (e.g., in athletes with high muscle mass relative to height). Further, BMI (weight) may increase over time not due to poor health behaviours, but medication side effects and hormonal changes associated with pregnancy or menopause (in women). This is another reason why it is conceptually and methodologically inappropriate to define BMI as a health behaviour, so any reference to BMI being a health behaviour should be deleted and/or edited throughout the manuscript. The previous reviewers had also made this comment but this has not been addressed by the authors.

Thank you for your comment. We have now deleted any reference to BMI as a behavior throughout the manuscript. For instance, the title is now changed to: "Educational patterns of health behavior and BMI…” to differentiate between BMI and the other variables related to health behavior. 

• The main outcome measures (health behaviours) were not very sensitive. For example, smoking was assessed as current, past or never – but did not include a measure of volume or duration, which is critical for estimating impacts on health. Pack years would have been a more sensitive measure. 

We understand the objection. While the question to assess smoking behavior was not very sensitive, current health guidelines do not establish any safe threshold for smoking behavior regarding both volume of consumption and frequency. We have made sure to include these aspects in our discussion section (see line 355-359). 

Also, participants were defined as physically active only if they reached the 150min/week threshold, but this obscures the ability to observe a dose-response relationship between education and physical activity. Further, the authors calculated physical activity minutes based on multiplying reported frequency and duration, but they estimated duration based on categorical data – this is not appropriate unless there is validation information for this?

Thank you for your relevant comment. The physical activity variable was coded to fit the units of current health recommendations, which is minutes per week. This process, which entailed multiplying the numerical values assigned to the answers to the questions regarding frequency and duration to obtain minutes per week has yet to be validated. The potential implications of this procedure have been now added in limitations in our discussion section (see line 345-349). 

Furthermore, while exploring a dose-response relationship between education and physical activity would have been of great interest, this was outside the scope of our study. The study focuses explicitly on cut-off points for each variable as stated in current health recommendations, which we have now pointed out more clearly.

• The decision to measure SES by education alone was confusing. The authors reasoned that this would be a good measure because it is stable, yet their study was longitudinal, so if their predictor is stable, what is the benefit of a longitudinal analyses, as this would make it less likely to observe changes in health behaviours as a result of changes in SES over time (because education does not change like occupation and income, which could affect the ability to ‘afford’ engaging in good health behaviours). The authors provided no hypotheses - is this what they expected to observe? This is indeed what they found, but this seems entirely predictable based on previous work and the nature/stability of this variable. What would have been more interesting is to examine interactions between education (stable variable) and occupation or income (unstable variable) which might help tease apart how these variables contribute to health behaviours. For example, education might be important for healthy eating, except when one loses their job and income – where buying healthy foods might be more difficult. Similarly, education might be associated with higher alcohol consumption (because drinking can be expensive and it is often seen as status symbol)- yet this association might decrease in those who lose their job or income.

Thank you for your relevant comment. After careful consideration, we have now replaced the term “socioeconomic status” and left it as education level, to convey that the primary focus of the paper is investigating the association between education level and the patterns in both BMI and the other three health behavior variables. Furthermore, we have expanded the discussion section (lines 367-379) to thoroughly address both the positive and negative aspects of using education as a time-invariant variable, as you have rightly pointed out. In addition, it is important to acknowledge that in our study, most respondents had completed their education by the initial measurement point. On the other hand, salary is subject to more fluctuations over time, which adds complexity to the analyses involving this variable. Additionally, it should be noted that health behaviors can also impact income changes, and we have recognized this aspect as well. Moreover, Norway exhibits relatively small salary differences compared to other countries (1) making it a less influential factor in explaining changes in compliance with health-related guidelines. It is worth mentioning that income data in the Tromsø Study waves remain self-reported and have not undergone validation. Consequently, education, which has been validated (2), remains the optimal choice within the context of our study.

We fully agree that all studies need to have a clear aim. We believe that our aim is relatively clear, and that we have fulfilled that aim with the present work. Since our work is, as noted, more descriptive in its form, we chose not to formulate a formal hypothesis. 

• Table 1 was confusing and hard to interpret because of the high number of independent groups that were presented across each health behaviour (and BMI is not a health behaviour). It would be more helpful and informative to classify each behaviour as getting better, worse or staying the same over time as a function of sex.

Thanks for your comment. The outcome of the MCA does precisely what the reviewer suggests, it presents the changes in the variables related to health behavior according to improvement, deteriorating or continuality, and was included separately for sex and age group. The reviewer mentioned that it could not be appreciated due to the low quality of the images. We apologize for that. However, the images were submitted according to the journal’s format requirement. 

• I was not able to read the figures because the resolution was not high enough. The text summarizing the results of the MCA analysis was also confusing – possibly because of the multiple analyses and largely descriptive nature of the findings.

All the images were submitted according to the journal’s format requirements. If figures do not have the desired resolution, we believe that must be due to a conversion made by the journal when converting into PDF. We encourage the reviewers to access the figures through the links in the PDF or by asking the journal editorial team.

• The discussion did not explain any of the mechanisms behind the observed associations, which really diminishes the contribution of the manuscript. The text on page 21, line 281 to 305, seems to go in circles and say no more than education is tied to health behaviours and this is generally maintained over time. How this is association may be explained by interactions with other risk factors was not discussed.

Thank you for your comment. We have now added potential explanations behind the observed associations in our discussion section to provide useful insights to the reader. 

• As the authors acknowledged, there is a high risk of selection bias and self-report bias, which undermines confidence in the data.

Thank you for pointing this out. The high risk of selection and self-report bias is discussed in lines 321-341. 

References: 

1. Kinge JM, Modalsli JH, Øverland S, Gjessing HK, Tollånes MC, Knudsen AK, et al. Association of Household Income With Life Expectancy and Cause-Specific Mortality in Norway, 2005-2015. Jama. 2019;321(19):1916-25.

2. Vo CQ, Samuelsen P-J, Sommerseth HL, Wisløff T, Wilsgaard T, Eggen AE. Validity of self-reported educational level in the Tromsø Study. Scandinavian journal of public health.0(0):14034948221088004.

---

## [Decision Letter · Decision Letter 2]

21 Nov 2023

Educational patterns of health behaviors and body mass index: a longitudinal multiple correspondence analysis of a middle-aged general population, 2007-2016

PONE-D-22-22752R2

Dear Dr. Ibarra-Sanchez,

We’re pleased to inform you that your manuscript has been judged scientifically suitable for publication and will be formally accepted for publication once it meets all outstanding technical requirements.

Kind regards,

Petri Böckerman

Academic Editor

PLOS ONE

Additional Editor Comments (optional):

I am happy with the paper.

Reviewers' comments:

Reviewer's Responses to Questions

**Comments to the Author**

1. If the authors have adequately addressed your comments raised in a previous round of review and you feel that this manuscript is now acceptable for publication, you may indicate that here to bypass the “Comments to the Author” section, enter your conflict of interest statement in the “Confidential to Editor” section, and submit your "Accept" recommendation.

Reviewer #5: All comments have been addressed

Reviewer #6: All comments have been addressed

2. Is the manuscript technically sound, and do the data support the conclusions?

Reviewer #5: Yes

Reviewer #6: Yes

3. Has the statistical analysis been performed appropriately and rigorously? 

Reviewer #5: Yes

Reviewer #6: I Don't Know

4. Have the authors made all data underlying the findings in their manuscript fully available?

Reviewer #5: Yes

Reviewer #6: Yes

5. Is the manuscript presented in an intelligible fashion and written in standard English?

Reviewer #5: Yes

Reviewer #6: Yes

6. Review Comments to the Author

Reviewer #5: (No Response)

Reviewer #6: Please see my comments to the editors. I am unable to provide any commentary on the merit of this paper. I have requested the editors to find another suitable reviewer for this purpose.

7. PLOS authors have the option to publish the peer review history of their article (what does this mean?). If published, this will include your full peer review and any attached files.

Reviewer #5: No

Reviewer #6: No

---

## [Editor Report · Acceptance letter]

24 Nov 2023

PONE-D-22-22752R2 

Educational patterns of health behaviors and body mass index: a longitudinal multiple correspondence analysis of a middle-aged general population, 2007-2016 

Dear Dr. Ibarra-Sanchez:

I'm pleased to inform you that your manuscript has been deemed suitable for publication in PLOS ONE. Congratulations! Your manuscript is now with our production department. 

Kind regards, 

on behalf of

Professor Petri Böckerman 

Academic Editor

PLOS ONE